# Dispersive-wave induced noise limits in miniature soliton microwave sources

Qi-Fan Yang [1,2], Qing-Xin Ji [1,2], Lue Wu [1,2], Boqiang Shen [1], Heming Wang [1], Chengying Bao[1], Zhiquan Yuan[1] & Kerry Vahala [1✉]

Compact, low-noise microwave sources are required throughout a wide range of application areas including frequency metrology, wireless-communications and airborne radar systems. And the photonic generation of microwaves using soliton microcombs offers a path towards integrated, low noise microwave signal sources. In these devices, a so called quiet-point of operation has been shown to reduce microwave frequency noise. Such operation decouples pump frequency noise from the soliton's motion by balancing the Raman self-frequency shift with dispersive-wave recoil. Here, we explore the limit of this noise suppression approach and reveal a fundamental noise mechanism associated with fluctuations of the dispersive wave frequency. At the same time, pump noise reduction by as much as 36 dB is demonstrated. This fundamental noise mechanism is expected to impact microwave noise (and pulse timing jitter) whenever solitons radiate into dispersive waves belonging to different spatial mode families.

---

[1] T. J. Watson Laboratory of Applied Physics, California Institute of Technology, Pasadena, CA, USA. [2]These authors contributed equally: Qi-Fan Yang, Qing-Xin Ji, Lue Wu. ✉email: vahala@caltech.edu

Soliton mode locking in optical microresonators is receiving intense interest for chip-scale integration of frequency comb systems[1]. Apart from frequency comb applications, the microwave signal produced by detection of the microcomb output is, itself, potentially important as a microwave signal source (see Fig. 1). However, mode locking of microcombs at microwave rates is challenging on account of their unfavorable pump power scaling with repetition rate[2]. Indeed, only ultra-high-Q discrete silica and crystalline devices were initially able to operate efficiently at microwave rates[2–5]. Nonetheless, the next generations of integrated ultra-high-Q resonators are emerging that both access the microwave-rate realm[6–8] and offer more complete integrated functionality[8,9]. Because of their superior phase noise performance compared to other miniature photonic microwave approaches[10–13], these devices are stimulating interest in miniature stand-alone soliton microwave sources.

While the fundamental limit of phase noise (and equivalently timing jitter) in the detected soliton pulse stream is induced by quantum fluctuations[14,15], in practice, phase noise is dominated by sources of a more technical origin that couple to the soliton motion in various ways. For example, the Raman self frequency shift in microcombs[16,17] provides a mechanism for transduction of changes in the detuning frequency (difference in the frequencies of the resonator mode being pumped and the pumping laser field) into the soliton repetition rate[18]. It does this by causing a frequency shift in the center frequency of the soliton spectrum (which has an overall sech$^2$ envelope) as the pump detuning frequency is varied. Group velocity dispersion then converts these spectral shifts into changes in the soliton round-trip propagation time and hence the repetition rate. The Raman process thereby couples any fluctuation of the resonator frequency (e.g., thermorefractive noise[19–22]) or the pump frequency into microwave phase noise. Dispersive waves can also induce a spectral center shift in the Kerr soliton[18,23]. Dispersive waves can emerge as a result of higher-order dispersion[23,24], supermodes[25], or when solitons radiate into resonator modes that do not belong to the soliton-forming mode family. And the spectral shift they induce can offset the Raman self shift. Indeed, when dispersive wave and Raman shifts are in balance, a *quiet* operating point is attained whereby coupling of detuning frequency fluctuations into the soliton repetition rate are greatly reduced[26].

Here, by investigating possible limits in application of the quiet operating point, we report the observation of a fundamental noise source in the soliton repetition rate. Referred to as spatiotemporal thermal noise, it originates from uncorrelated thermal fluctuations between distinct transverse modes of the microresonator, and can couple into the soliton repetition rate through the formation of a dispersive wave. Theory and experiment show that the spatiotemporal thermal noise imposes a considerable limitation on the repetition rate stability of soliton microcombs emitting dispersive waves into spatial mode families, that are distinct from the soliton-forming mode family. Beyond the study of the dispersive-wave noise, a convenient way to operate the soliton microwave source at the quiet point while also disciplining it to an external reference, such as a clock, is demonstrated.

## Results

**Soliton generation in silica microresonators.** A silica disk microresonator with intrinsic $Q$ factor exceeding 300 million and free-spectral-range (FSR) around 15 GHz is used in the study[27,28]. The microresonator is packaged with active temperature stabilization[29] and operated under an acoustic shield to block environmental perturbations (Fig. 2a, b). By continuously pumping the resonator with an amplified fiber laser, bright soliton pulses are generated, which are further stabilized by servo control of the pump laser frequency with respect to the average soliton power[30]. The residual error in the feedback loop is monitored by an electrical spectrum analyzer. The soliton beatnote is photodetected and characterized using a phase noise analyzer and a frequency counter. The beatnote of the soliton microcomb shows a 15.2 GHz repetition rate (see Fig. 1). Its phase noise exhibits a smooth spectral shape across a wide frequency range as a result of isolation provided by the package and acoustical shield (Fig. 2b). As a benchmark of the stability, the fractional Allan deviation of the beatnote is plotted in Fig. 2c and reaches $5.7 \times 10^{-11}$ at 50 ms averaging time.

**Quiet point operation.** Plotted in Fig. 3a is a representative optical spectrum of the soliton microcomb, showing its characteristic sech$^2$ spectral envelope. Dispersive waves (the spectral spurs on the envelope) also appear in the spectrum and result

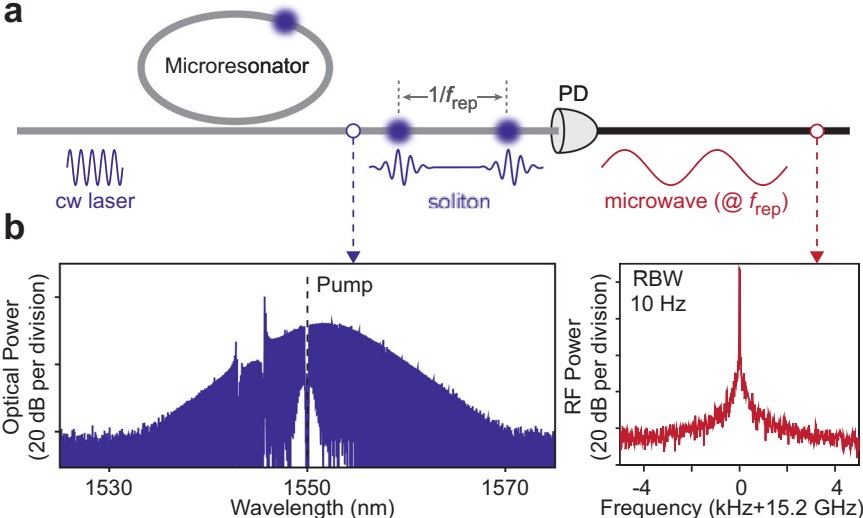

**Fig. 1 Soliton microcombs as microwave signal sources. a** Apparatus for microwave signal generation using a soliton microcomb. A microresonator pumped by a continuous-wave (cw) laser emits a repetitive soliton pulse train that is directed into a photodetector (PD) to produce a signal current. **b** Representative optical spectrum of a soliton microcomb with 15.2 GHz repetition rate (left panel). The pump (black dashed line) has been attenuated by an optical notch filter. The right panel shows the corresponding microwave-rate beat signal with resolution bandwidth (RBW) of 10 Hz.

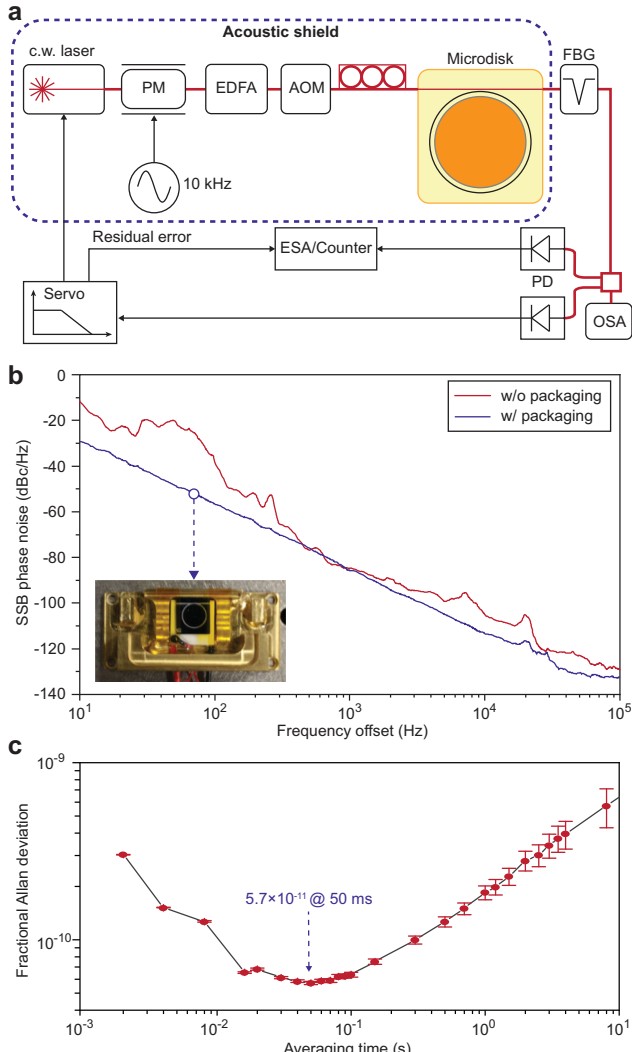

**Fig. 2 Experimental setup and preliminary microwave signal characterization. a** Experiment setup for soliton generation. PM phase modulator, EDFA erbium-doped-fiber-amplifier, AOM acousto-optic modulator, FBG fiber-Bragg-grating notch filter, PD photodetector, OSA optical spectral analyzer, ESA electrical spectral analyzer. **b** Typical single-sideband (SSB) phase noise spectrum of detected soliton pulse stream (scaled to 15.2 GHz) obtained using packaged/unpackaged microresonators. Inset: photo of a packaged microresonator. **c** Fractional Allan deviation of soliton pulse rate. The errorbar indicates standard deviation.

from frequency degeneracy between comb lines and other transverse modes, that do not belong to the soliton forming mode family[3,18,26,31]. It is noted that the spectral envelope center of the soliton is offset from the pump frequency. This is caused by the cumulative effect of the Raman-induced soliton-self-frequency-shift (SSFS) $\Omega_{Raman}$[16–18] and dispersive-wave induced spectral recoil $\Omega_{recoil}$[26,31,32]. The soliton repetition rate, $\omega_{rep}$, is related to these frequency shifts by[18,26]

$$\omega_{rep} = D_1 + \frac{D_2}{D_1}(\Omega_{Raman} + \Omega_{recoil}), \qquad (1)$$

where $D_1/2\pi$ is the FSR and $D_2$ is proportional to the group velocity dispersion (GVD) of the soliton-forming mode family[2,33]. Therefore, through the respective dependence of $\Omega_{Raman}$ and $\Omega_{recoil}$ on the detuning frequency $\delta\omega = \omega_o - \omega_P$ ($\omega_o$ is the frequency of the cold cavity mode being pumped by optical

field at frequency $\omega_P$), the soliton repetition rate becomes a function of the detuning frequency. As reported in previous literature, noise in $\omega_P$ often plays a dominant role in causing fluctuations in $\delta\omega$, and subsequently, by way of Eq. (1), also in $\omega_{rep}$[26,34,35]. However, it has also been shown that interplay between Raman SSFS and dispersive-wave induced spectral recoil can be used to suppress this noise transfer[26,35]. Along these lines, Fig. 3b is the measured dependence of soliton repetition rate on detuning $\delta\omega/2\pi$, and shows a parabolic-like trend instead of a monotonic trend. The slope, $\beta = \partial\omega_{rep}/\partial\delta\omega$, vanishes at around 11.5 MHz detuning, corresponding to the quiet point of operation where dispersive wave and Raman induced shifts are in balance. Here, the detuning $\delta\omega$ is calculated based on Eq. (23) in "Methods" section. By operating the soliton microcomb near this quiet point, the contribution of detuning noise to the soliton repetition rate noise can be reduced[26,35].

To actively monitor the degree to which the detuning noise contribution is suppressed through quiet point operation, we modulate the phase of the pump laser at 10 kHz to create a large spike in the detuning noise spectrum. This induces calibration tones in the vicinity of the soliton beatnote[35], as shown in Fig. 3c. Measured phase noise spectra of the detected soliton microwave signal along with the power of calibration tone (see colored triangle points) are plotted in Fig. 3d for different detuning frequencies. As an aside, the pronounced bump around 20 kHz in the phase noise spectrum is caused by the piezoelectric tuning bandwidth of the pump laser. Away from the quiet point, the phase noise is largest and is found to follow the spectral profile of the detuning noise, which is extracted from the residual error signal in the locking loop. The contribution of the detuning noise can be scaled based on the power of calibration tone to determine its contribution in each measurement. At the quiet point, 36 dB of noise suppression is measured using the calibration tone. And the corresponding inferred detuning noise contribution (dashed red spectrum in Fig. 3d) is below the actual measured noise spectrum at the quiet point (purple spectrum in Fig. 3d). This indicates that another noise source is limiting the phase noise at the quiet point. As one possible source of this limit, pump intensity noise could also couple into the soliton repetition rate through the combined effect of Kerr and Raman nonlinearity[34,35]. However, its contribution (see dashed gray curve in figure) is evaluated in the "Methods" section and appears to be negligible in this measurement. Figure 3e gives a comparison of the measured phase noise reduction (referenced to the highest phase noise trace) versus the reduction inferred by the calibration tone. A clear saturation in the measured noise reduction near the quiet point is shown at several different offset frequencies, suggesting again that a source of noise is present. The saturation is stronger at lower offset frequencies indicating that the noise mechanism is larger at lower frequencies (see Fig. 3d). As an aside, the quiet-point-induced phase noise reduction is also slightly higher than indicated by the calibration tone for lower noise suppression levels (when measured at 500 Hz and 1 kHz offsets). This could result from possible instrument calibration error associated with calibration using a 10 kHz tone.

**Dispersive-wave induced noise.** Prior analysis of fundamental sources of repetition rate noise assume that the soliton is formed and couples solely within a single transverse mode family. However, the practical need for higher $Q$ resonators favors larger resonator cross-section to minimize the impact of interface and sidewall roughness[36]. Typically, several transverse modes besides the soliton forming mode exist in the microresonator. And when longitudinal modes in these other families experience near degeneracy with a mode in the soliton, the soliton radiates power

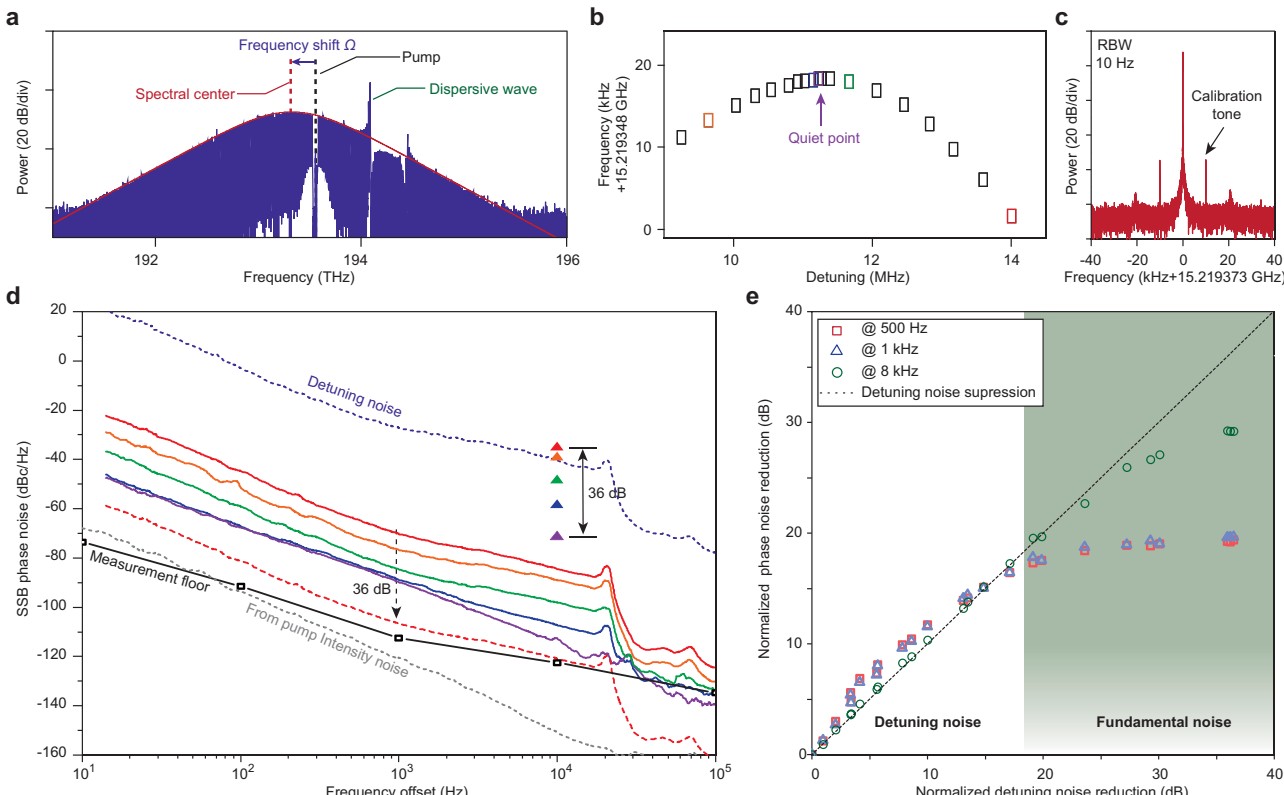

**Fig. 3 Noise spectra near and away from the quiet point. a** Soliton optical spectrum showing spectral envelope (red solid line), the attenuated pump (black dashed line) and a strong dispersive wave. The spectral center of the soliton (red dashed line) is shifted in frequency relative to the pump frequency. **b** Measured soliton repetition rate versus laser-cavity detuning ($\delta\omega/2\pi$), where the existence of a quiet point is revealed. **c** Electrical spectrum showing soliton repetition rate. Two sidebands at 10 kHz offset frequency are induced by phase modulation of the pump and are used to calibrate the contribution of detuning noise. **d** Single-sideband (SSB) soliton microwave phase noise (solid curves) and calibration tone power (triangles) at different detuning frequencies (indicated by color in accordance with **b**). The optical detuning noise is the blue dotted line. At the quiet operating point, its calibration-inferred contribution to microwave noise is the dashed red curve. Noise induced by the pump intensity fluctuation (gray dotted line) is also plotted. The phase noise analyzer instrumental noise floor is shown as the black line. **e** Plot of actual noise suppression versus calibration tone suppression at several offset frequencies. The dashed line indicates the expected phase noise suppression if detuning noise is dominant.

creating a dispersive wave (Fig. 4a)[3,18,26,31]. The radiative power depends strongly upon the degree of resonance as determined by $\Delta\omega$ (the frequency difference between the two modes), $\Delta\omega_r$ (the frequency difference between the soliton comb line and the soliton-forming mode with index $r$), and $\kappa_B$ (the optical loss rate of the dispersive wave mode). The relationship between these difference frequencies is illustrated in Fig. 4a. The radiated power causes a frequency recoil, $\Omega_{recoil}$, in the soliton spectral center relative to the pump frequency which takes the form[26]

$$\Omega_{recoil} \propto \frac{1}{(\Delta\omega')^2[(\Delta\omega_r - \Delta\omega')^2 + \frac{\kappa_B^2}{4}]}, \quad (2)$$

where $\Delta\omega'$ is the frequency difference between the partially hybridized crossing mode and the soliton mode, denoted by $\Delta\omega' = \Delta\omega/2 + \sqrt{\Delta\omega^2/4 + G^2}$ (where $G$ is the coupling strength between the soliton and crossing mode). $\Delta\omega_r$ is determined by both detuning $\delta\omega$ and recoil (and thereby $\Delta\omega$). And this equation provides a way for fluctuations in $\delta\omega$ and $\Delta\omega$ to impact the soliton repetition rate. Specifically, the resulting fluctuations in $\Omega_{recoil}$ cause spectral center fluctuations of the soliton that randomly vary its round trip time as a result of second order dispersion. The physical process steps involved in this noise transduction mechanism are depicted in Fig. 4b. A transduction factor $\alpha \equiv \partial\omega_{rep}/\partial\Delta\omega$ relating the repetition rate to changes in $\Delta\omega$ is defined and noted in the figure. For comparison, the process steps involved in the transduction of detuning noise into

repetition rate changes ($\beta$ factor defined earlier) are also provided. As noted earlier, detuning noise can be quieted through interference between the pathways indicated in Fig. 4b, one of which uses portions of the dispersive wave recoil process.

To identify the mode families that constitute the soliton microcomb and the dispersive wave in the experiment, we perform mode family dispersion spectroscopy using a scanning external-cavity-diode-laser (calibrated by a separate Mach-Zehnder interferometer), as shown in Fig. 4c. Comparing the measurement with numerical modeling of the modal dispersion, the mode family that gives rise to the strong dispersive wave in Fig. 3a is determined to belong to the $TM_4$ mode family, while the soliton is formed on the $TM_0$ mode family. Their $Q$ factors are also measured, as shown in Fig. 4d.

**Thermal noise in the dispersive wave.** Fluctuations associated with thermal equilibrium result in spatial and temporal variations of temperature in the microresonator[19–22,37]. Such temperature fluctuations, characterized by a spectral density $S_{\delta T}$ of the modal temperature fluctuations, induce frequency fluctuations $\delta D_1$ in the resonator FSR through the thermo-optic effect. In turn, this induces fluctuations in the soliton repetition rate that are characterized by the spectral density $S_{\delta D_1}$,

$$S_{\delta D_1} = \frac{n_T^2}{n_o^2} D_1^2 S_{\overline{\delta T}}, \quad (3)$$

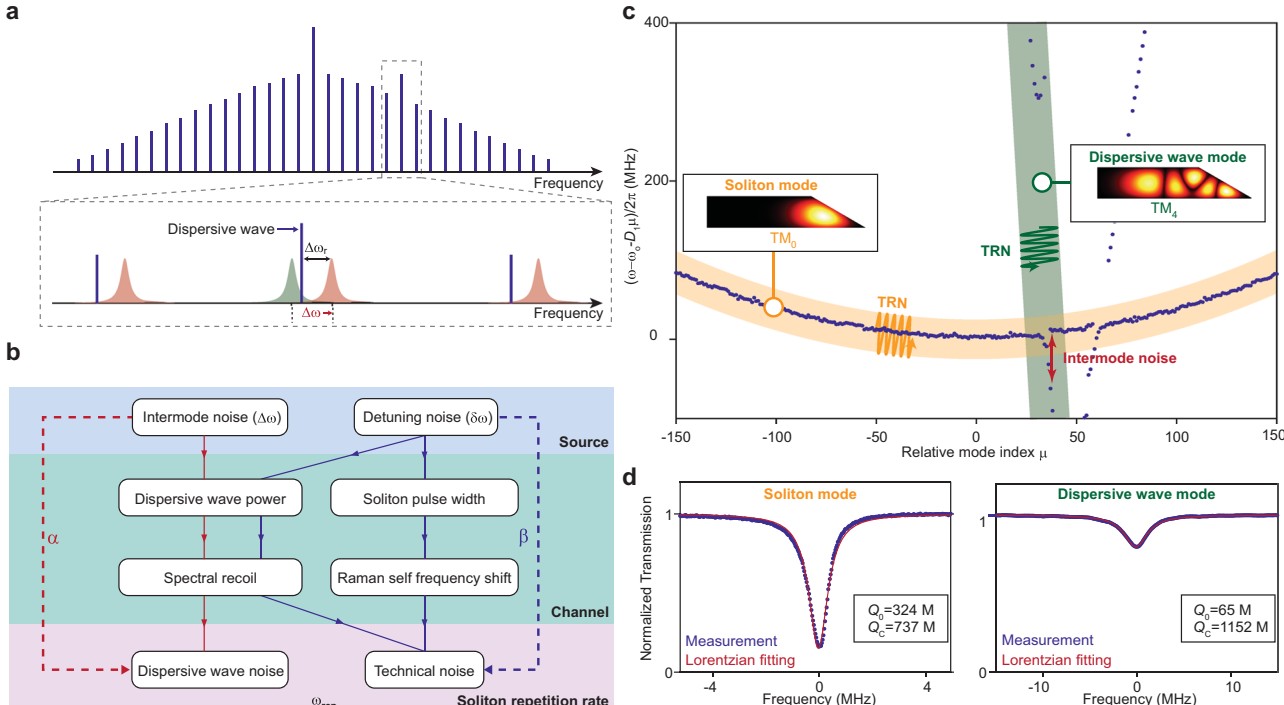

**Fig. 4 Concept of dispersive-wave-induced noise and identification of mode families. a** Spectral relationship of soliton spectrum to dispersive wave forming mode. Blue lines indicate soliton spectral lines. Red and green shaded regions denote soliton-forming and dispersive-wave resonator modes, respectively. **b** Left side: physical steps involved in coupling fluctuations in $\Delta\omega$ (intermode noise) into soliton repetition rate. Right side: physical steps involved in coupling fluctuations in $\delta\omega$ (detuning noise) into soliton repetition rate. Noise sources (top) are transduced ($\alpha$ and $\beta$ coupling channels) into the soliton repetition rate (bottom). Detuning noise results mainly from the pumping laser noise contributing to $\delta\omega$ and thereby causes technical noise in the soliton repetition rate. Intermode noise results from fundamental thermo-refractive noise of the dispersive wave and soliton mode frequencies contributing to $\Delta\omega$. **c** Measured mode family dispersion of the microresonator. Numerically simulated cross sections of soliton ($TM_0$) and dispersive wave ($TM_4$) modes are plotted and identified with the corresponding frequency branches. Orange and green bands (and wavy lines) are suggestive (and highly magnified) fluctuations induced by thermo-refractive noise (TRN). **d** Measured transmission spectra of soliton and dispersive wave resonator modes. The intrinsic ($Q_O$) and coupling ($Q_C$) Q-factors are extracted by fitting the Lorentzian lineshapes and transmission minima.

with $n_T$ the thermo-optic coefficient and $n_o$ the refractive index of the mode. This noise contribution to the soliton repetition rate, and that induced by quantum vacuum fluctuations[14,15], are found to be much smaller than the measured noise in Fig. 3d. However, as now shown, thermorefractive noise (TRN) induced in the modes participating in dispersive-wave emission can be a major source of repetition rate noise.

From the analysis in the previous section, noise in relative frequency, $\Delta\omega$, will couple to the repetition rate through the parameter $\alpha$. The TRN induced noise in $\Delta\omega$ is given by the following spectral density (see" Methods" section),

$$S_{\Delta\omega} = \frac{n_T^2}{n_o^2}\omega_o^2(S_{\delta T_S} + S_{\delta T_D} - 2R\sqrt{S_{\delta T_S}S_{\delta T_D}}), \qquad (4)$$

where $\delta T_S$ and $\delta T_D$ give temperature fluctuations of mode volumes associated with the soliton and dispersive-wave modes involved in the definition of $\Delta\omega$. $R$ is a frequency dependent function discussed in the "Methods" section that accounts for correlation between the fluctuations $\delta T_S$ and $\delta T_D$. This correlation can be modeled using the finite-element-method (FEM) and the fluctuation-dissipation theorem (FDT)[21,22,37]. Simulation results for different pairs of transverse modes are plotted in Fig. 5a. On account of thermal diffusivity, the function $R$ decreases rapidly with increasing frequency, so that beyond a thermal-limited rate the temperature fluctuations of the two modes become uncorrelated. When this happens, the value of $S_{\Delta\omega}$ exceeds $S_{\delta D_1}$ by several orders since it reflects temperature

fluctuations in absolute (as opposed to relative) optical frequencies.

In order to test the numerical results and parameters used to simulate these thermally-related quantities[21,22], we measured the TRN of the soliton mode. The frequency fluctuations of the mode were tracked by Pound–Drever–Hall (PDH) locking a fiber laser to a cavity resonance. The locked laser frequency is then measured using an optical frequency discriminator as described in the "Methods" section. The measured single-sideband TRN is plotted in Fig. 5b, and is in good agreement with the simulation. The calculated phase noise of the intermode TRN using the soliton mode and dispersive wave mode is also plotted for comparison. Suppression of intermode TRN is apparent at low-offset frequencies relative to the single mode TRN. However, at higher offset frequencies (above ~1 kHz), the intermode TRN becomes the summation of TRN contributions belonging to each mode. The TRN of the FSR is also shown for comparison. Notice that despite the improved correlation of the intermode TRN at lower offset frequencies, it still dominates the microwave phase noise measured in Fig. 3d, e. This happens because the TRN noise rises very rapidly as offset frequency decreases, even overcoming the improving correlation of TRN between the dispersive wave mode and soliton forming mode.

An additional measurement of soliton microwave phase noise was performed except using PDH locking of the pump laser to the resonator as opposed to servo control using soliton power. Under these conditions, the pump frequency tracks the cavity resonance thereby suppressing its technical noise contribution to the soliton

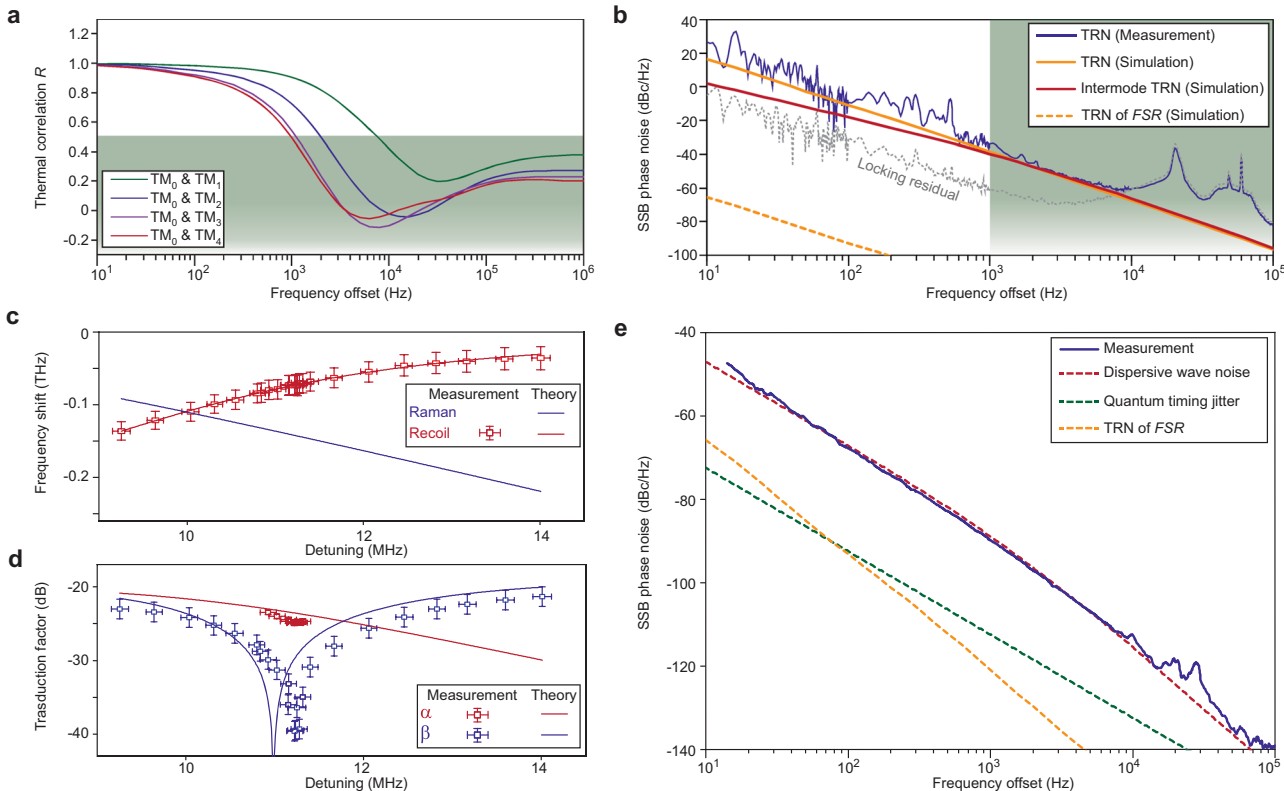

**Fig. 5 Intermode thermal noise (between dispersive wave and soliton modes) and its impact on soliton repetition rate. a** Simulated temperature correlation $R$ between transverse mode volumes versus frequency of thermal fluctuation. Specific transverse mode pairs are indicated in the legend. Green region corresponds to $R < 0.5$. **b** Measured and simulated single-sideband (SSB) TRN of a $TM_0$ mode. The simulated intermode TRN between $TM_0$ and $TM_4$ is also displayed. Green region corresponds to $R < 0.5$ in **a**. **c** Contribution of Raman SSFS and dispersive recoil to total spectral center frequency shift of the soliton. The error bar indicates standard deviation, and is contributed from fitting of the lineshape. **d** Measured and calculated noise transduction factors. The error bar indicates standard deviation. The error in detuning is contributed by the lineshape fitting, while the transduction factor error comes from the signal analyzer. **e** Measured phase noise at maximum quiet point suppression and calculated dispersive-wave induced noise originating from intermode TRN. Quantum timing jitter and thermorefractive noise (TRN) of the FSR are also plotted for comparison.

phase noise. As expected the measured noise spectrum showed a limitation consistent with the dispersive wave noise (see Supplementary note II).

A summary of noise contributions to the soliton repetition rate yields

$$S_{\omega_{rep}}(f) = \alpha^2 S_{\Delta\omega} + S_{\delta D_1} + S_Q + \beta^2 S_{\delta\omega} + S_P, \quad (5)$$

where $S_Q$ is the quantum noise limit[14], and $S_P$ is noise transferred from intensity noise of the pump laser. To evaluate the noise transduction factors, experimental results are fitted with theory based on the Lugiato–Lefever equation (see "Methods" section). The Raman frequency shift and dispersive-wave recoil are plotted in Fig. 5c, where the error bar is the standard deviation contributed from fitting of the soliton spectral envelope. The frequency recoil in Eq. (2) is fitted in the same graph to evaluate mode coupling coefficients. Noise transduction factors $\alpha$ and $\beta$ are then calculated and plotted in Fig. 5d together with the measured results. Figure 5e shows both measured and calculated phase noise of the soliton repetition rate while operating at the quiet point. Excellent agreement with the predicted intermode TRN induced noise is obtained by setting $\alpha = -24.5$ dB (measurement value), which is close to the theoretical value $\alpha = -23.6$ dB. Other fundamental noise contributions are also plotted, but are not limiting factors in the current measurement[14,19–22].

**External reference locking at the quiet point**. Most signal sources provide a feature that allows the oscillator frequency to be conveniently locked to an external reference such as a clock so as to provide long term frequency stability[38]. In the present device, there is a straightforward way to achieve this locking that also provides fine tuning control of the microwave frequency near the quiet point. As a proof of concept, instead of servo controlling the soliton system by controlling the soliton power[30], we lock the soliton repetition rate to a high-performance electrical signal generator by servo controlling the optical pump frequency. The resulting soliton beatnote is shown in Fig. 6a, and can track the frequency of the microwave source over a 30 kHz range to achieve fine tuning control. This range is likely determined by the soliton existence range, which is, in turn, determined by the pump laser power[33]. At the same time, the soliton microwave phase noise, shown in Fig. 6b, is disciplined to the reference oscillator within the servo locking bandwidth. The peak around 10 kHz is induced by the servo locking bandwidth. At high-offset frequencies, the soliton phase noise outperforms the electrical oscillator (a Keysight PSG) by up to 20 dB. A variation in noise performance with fine tuning is apparent with the best performance corresponding to operation near the quiet point.

**Discussion**. In conclusion, we demonstrated a low-noise 15 GHz oscillator based on soliton microcombs. The measured phase noise of $-90$ dBc Hz$^{-1}$ at 1 kHz and $-140$ dBc Hz$^{-1}$ at 100 kHz offset frequencies is a record low among existing photonic-chip-

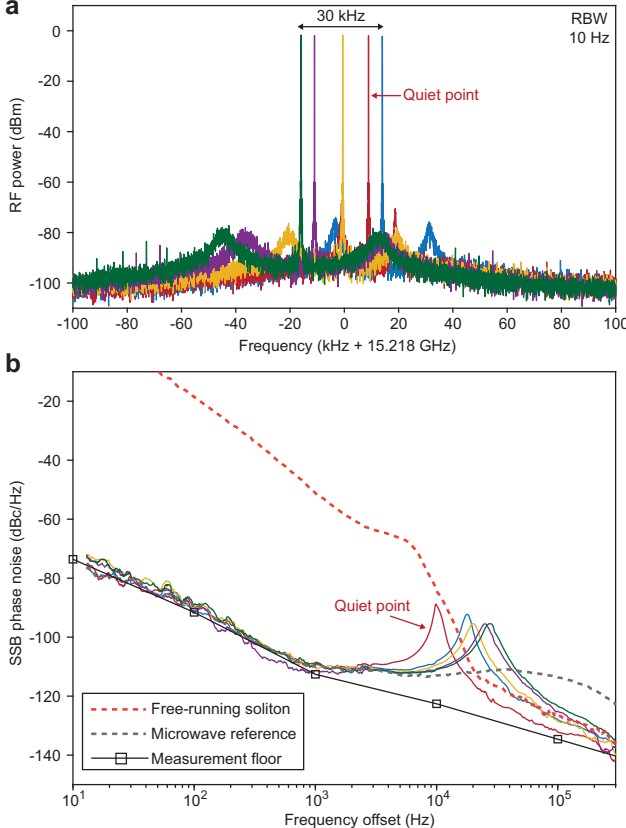

**Fig. 6 Soliton repetition rate disciplined to an external microwave source. a** Electrical beatnote of locked soliton repetition rate showing fine tuning control of the repetition rate. The resolution bandwidth (RBW) is 10 Hz. **b** Phase noise of free-running (dashed red line) and disciplined soliton microcomb (indicated by color in accordance with panel **a**). The trace measured near the quiet point is indicated. The phase noise of the microwave reference source is also displayed (gray dashed line).

based microwave sources[7,8,10–13] (scaled to 15 GHz). A comparison of miniature photonic-based microwave oscillators is included in Table. 1. The low noise performance was obtained by operation near the soliton mode locking quiet point[26], where technical noise suppression as large as 36 dB was measured. Discipline of the soliton microwave source to an external microwave reference was also demonstrated, which could be useful in combination with miniaturized optical clocks[38].

A fundamental noise mechanism associated with dispersive waves belonging to non soliton-forming mode families was also identified and theoretically modeled. Since dispersive waves induced by distinct transverse modes are ubiquitous across many soliton microcomb systems[2,7,33,39,40], this noise mechanism is expected to appear in other soliton microwave systems. Nonetheless, several methods can be implemented to mitigate this noise. First, use of dispersive waves within the same longitudinal mode family (as formed by higher order dispersion[23,24,41,42]) could be investigated. In this case, better overlap of the dispersive-wave modal profile with the soliton mode would be expected to reduce the dispersive wave noise. Also, increasing the modal volume, reducing the thermo-optic coefficient or moving to cryogenic temperatures[43] could also greatly enhance the thermal stability of the microresonator[4,44]. Such techniques might ultimately endow these photonic microwave sources with quantum-limited performance[14].

## Methods

**Experimental details.** The resonant frequencies of the modes are measured by scanning an external cavity diode laser across a broad wavelength span (1520–1630 nm in this measurement). The laser scan is precisely measured by a radio-frequency calibrated Mach–Zehnder interferometer[2,45]. The resonant frequency at mode index $\mu$ is expanded up to the second order with respect to the mode number $\mu$,

$$\omega_\mu = \omega_o + D_1\mu + \frac{1}{2}D_2\mu^2 + \mathcal{O}(\mu^3). \qquad (6)$$

From the measurement, the parameters of the soliton mode family are: $D_{1S}/(2\pi) = 15.21857$ GHz, $D_{2S}/(2\pi) = 7.5$ kHz, and the parameters of the dispersive wave mode mode family are: $D_{1D}/(2\pi) = 15.17479$ GHz, $D_{2D}/(2\pi) = 7.5$ kHz.

The soliton microcomb is amplified to around 5 mW using an EDFA before coupling into the high-speed photodetector. The phase noise of soliton repetition

### Table 1 Comparison of miniature reported photonic-based microwave oscillators.

| | | | Phase noise | | |
| | | | SSB phase noise (dBc/Hz, scaled to 15.2 GHz) | | Refs. # |
| **Material** | **Configuration** | **Carrier freq. (GHz)** | **1 kHz** | **100 kHz** | |
| SiO$_2$ (this work) | Bright soliton | 15.2 | −90 | −140 | |
| MgF$_2$ | Bright soliton | 14.1 | −121 | −155 | 35 |
| Si$_3$N$_4$ | Bright soliton | 19.6 | −82 | −132 | 7 |
| Si$_3$N$_4$ | Dark soliton | 5.4 | −76 | −131 | 8 |
| SiO$_2$ | SBS | 21.7 | −71 | −113 | 10 |
| Si$_3$N$_4$ | SBS | 21.8 | −55 | −102 | 13 |
| Chalcogenide-on-silicon | SBS | 40.0 | −93 | −110 | 52 |

| | | | Relative Allan deviation | | |
| **Material** | **Configuration** | **Min. Adev. gate time (ms)** | **Min. Adev.** | **Adev. @ 1 s** | **Refs. #** |
| SiO$_2$ (this work) | Bright soliton | 50 | $6 \times 10^{-11}$ | $2 \times 10^{-10}$ | |
| MgF$_2$ | Bright soliton | 200 | $5 \times 10^{-12}$ | $10^{-11}$ | 4 |
| Si$_3$N$_4$ | Bright soliton | 1 | $3 \times 10^{-9}$ | $8 \times 10^{-8}$ | 7 |
| SiO$_2$ | SBS | 20 | $10^{-9}$ | $10^{-8}$ | 10 |

Performance of an on-chip silica bright soliton microcomb (this work), a Crystalline bright soliton microcomb[4,35], a Si$_3$N$_4$ bright soliton microcomb[7], a Si$_3$N$_4$ dark soliton microcomb[8], a SiO$_2$ Brillouin laser[10], a Si$_3$N$_4$ Brillouin laser[13], and a hybrid electric-Brillouin oscillator[52].

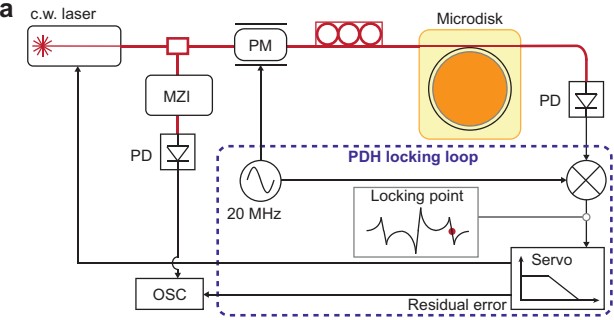

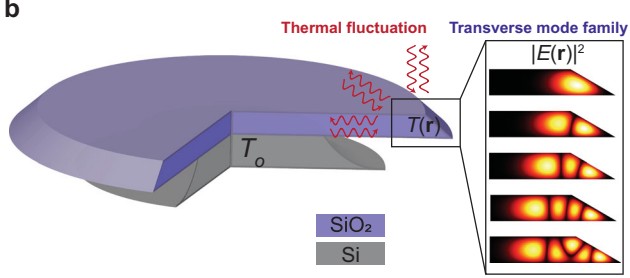

**Fig. 7 TRN measurement and resonator geometry. a** TRN measurement setup. MZI Mach–Zehnder interferometer, OSC oscilloscope, PD photodiode, PM phase modulator. **b** Cross-sectional view of the microresonator. Multiple transverse modes are supported in the suspended wedge-shaped whispering gallery. The surrounding silica, silicon, and air serve as a heat reservoirs. The ambient temperature is denoted by $T_o$.

rate is measured using a Rohde–Schwarz FSUP26 phase noise analyzer with cross-correlation function. The soliton beatnote is down-mixed with a high-performance electrical oscillator (Agilent E8257D PSG analog signal generator) before sending into a frequency counter for the measurement of Allan deviation.

The detuning noise is obtained by monitoring the residual error signal of the locking loop. The transduction factor (slope) between error signal and pump-cavity detuning is calibrated by mapping the error signal with respect to different soliton operation points. No obvious dependence of detuning noise on specific soliton operation point is observed, indicating that the detuning noise is primarily determined by the pump laser and the servo.

The TRN measurement setup is shown in Fig. 7a. A fiber laser is locked to a soliton forming mode using the Pound–Drever–Hall (PDH) locking technique. To mitigate the thermo-optic locking effect[46–48], the power launched into the microresonator is reduced. Also, the locking point is set to a sideband of the PDH error signal to reduce the power coupled into the microresonator. The TRN is then extracted by monitoring the laser frequency in real time using a Mach–Zehnder inteferometer as an optical frequency discriminator.

The error of the results are contributed from instrumental and fitting errors. In the phase noise measurement, the signal analyzer contributes an uncertainty of <1 dB for 100 Hz to 10 MHz offset and <3 dB for 1–100 Hz and 10–30 MHz offset. In the detuning noise measurement, the measurement uncertainty is <3 dB. Fitting uncertainty is evaluated using nonlinear regression.

**Thermal noise theory**. In this section, we derive the spectral density of modal temperature fluctuation based on the fluctuation-dissipation theorem (FDT)[21,37,49,50]. As shown in Fig. 7b, the microresonator exists in a heat reservoir with temperature $T_o$. The temperature deviation from thermal equilibrium follows the heat equation

$$\rho C \frac{\partial \delta T}{\partial t} - k\nabla^2 \delta T = \frac{\partial \delta Q}{\partial t} = T_o \frac{\partial \delta S}{\partial t}, \quad (7)$$

where $\rho$, $C$, and $k$ are respectively the material mass density, heat capacity and thermal conductivity. $\delta Q$ and $\delta S$ are the local fluctuation of heat and entropy. First we study the fluctuation of the optical-mode-weighted average temperature, which takes the form

$$\overline{\delta T} = \int \delta T q(\mathbf{r}) d^3 \mathbf{r}. \quad (8)$$

Here the density $q(\mathbf{r})$ represents the normalized distribution of the electrical field

intensity (Fig. 7b), which can be written as

$$q(\mathbf{r}) = \frac{|E(\mathbf{r})|^2}{\int |E(\mathbf{r})|^2 d^3 \mathbf{r}}. \quad (9)$$

As described in previous literature[21,37], to properly formulate the FDT a periodic entropy injection is applied onto the system such that

$$\delta S = F_o \cos(2\pi f t) q(\mathbf{r}). \quad (10)$$

The resulting time-averaged dissipation power yields

$$W_{\text{diss}} = \int \frac{k}{T_o} < (\nabla \delta T)^2 > d^3 \mathbf{r}, \quad (11)$$

where <> denotes time-averaging, which, according to the FDT, gives the single-sideband power spectral density (PSD) of $\overline{\delta T}$,

$$S_{\overline{\delta T}}(f) = \frac{\hbar W_{\text{diss}}}{\pi f F_o^2} \coth\left(\frac{\pi \hbar f}{k_B T_o}\right), \quad (12)$$

with $k_B$ Boltzmann's constant.

This approach can be extended to reveal the PSD of temperature difference between two optical modes, $S_{\Delta T}$, by introducing the difference in field intensity distributions as follows,

$$q(\mathbf{r}) = q_1(\mathbf{r}) - q_2(\mathbf{r}) = \frac{|E_1(\mathbf{r})|^2}{\int |E_1(\mathbf{r})|^2 d^3 \mathbf{r}} - \frac{|E_2(\mathbf{r})|^2}{\int |E_2(\mathbf{r})|^2 d^3 \mathbf{r}}, \quad (13)$$

where $E_1$ and $E_2$ represent the respective electrical fields of the two modes. From these results, the spectral density of the difference in modal-weighted temperatures is given by,

$$\begin{aligned} S_{\Delta T}(\omega) &= \mathcal{F}\{<[\overline{\delta T_1}(t) - \overline{\delta T_2}(t)][\overline{\delta T_1}(t+\tau) - \overline{\delta T_2}(t+\tau)]>\}(\omega) \\ &= S_{\overline{\delta T_1}}(\omega) + S_{\overline{\delta T_2}}(\omega) - 2\text{Real}\{\mathcal{F}[<\overline{\delta T_1}(t)\overline{\delta T_2}(t+\tau)>](\omega)\}, \end{aligned} \quad (14)$$

where Fourier transformation is denoted by $\mathcal{F}$. With the definition,

$$R(\omega) = \frac{\text{Real}\{\mathcal{F}[<\overline{\delta T_1}(t)\overline{\delta T_2}(t+\tau)>](\omega)\}}{\sqrt{S_{\overline{\delta T_1}} S_{\overline{\delta T_2}}}}. \quad (15)$$

Equation (14) gives Eq. (4) in the main text when the thermo-optic properties are taken into account.

In practice, the energy dissipation $W_{\text{diss}}$ can be acquired by Fourier transformation of Eq. (7) with respect to $t$, yielding

$$i\omega\rho C\mathcal{F}\{\delta T\} - k\nabla^2 \mathcal{F}\{\delta T\} = i\omega T_o F_o q(\mathbf{r}), \quad (16)$$

where $\omega = 2\pi f$ and only positive frequency components of $\delta S$ are considered. An FEM solver (COMSOL multiphysics in this work) can be used for simulations using the above equation. Critical parameters used in the simulation of thermal properties are: density $\rho = 2.2 \times 10^3$ kg m$^{-3}$, heat capacity $C = 740$ J kg$^{-1}$ K$^{-1}$, thermal conductivity $k = 1.38$ W m$^{-1}$ K$^{-1}$, thermorefractive index $n_T = 1.2 \times 10^{-5}$ K$^{-1}$, and ambient temperature of 300 K. The silica resonator has 22 mm radius and 8 μm thickness, supported by a silicon pillar with 140 μm in undercut. The wedge angle is 30°.

**Theory of noise transduction**. In this section, we overview the theoretical analysis used to determine the noise transduction factors $\alpha$ and $\beta$. The following pair of Lugiato–Lefever equations[26,33,51] are utilized to predict the dynamics of the soliton field with another transverse mode family field coupled to it so as to provide dispersive wave radiation,

$$\frac{\partial E_S}{\partial t} = \left(-\frac{\kappa_S}{2} - i\delta\omega + i\frac{D_{2S}}{2}\frac{\partial^2}{\partial\phi^2} + ig_S|E_S|^2 + i\gamma_S\frac{\partial|E_S|^2}{\partial\phi}\right)E_S + f_o + iGE_D, \quad (17)$$

$$\begin{aligned} \frac{\partial E_D}{\partial t} = &\left[-\frac{\kappa_D}{2} - i(\delta\omega + \Delta\omega_o - i\Delta D\frac{\partial}{\partial\phi}) + i\frac{D_{2D}}{2}\frac{\partial^2}{\partial\phi^2}\right. \\ &\left. + ig_D|E_D|^2 + i\gamma_D\frac{\partial|E_D|^2}{\partial\phi}\right]E_D + iGE_S, \end{aligned} \quad (18)$$

where $E_S$ ($E_D$) is the slowly varying field envelope (photon number normalization) for the soliton (dispersive wave), $\kappa_{S,D}$ are the corresponding energy decay rates, $\delta\omega$ is the pump-cavity frequency detuning, $g_{S,D} \equiv \hbar\omega_o^2 n_2 D_{1S,D}/2\pi n_o A_{\text{eff}}$ is the nonlinear coupling coefficient with $A_{\text{eff}}$ the effective nonlinear mode area, $\gamma_{S,D} \equiv g_{S,D}D_{1S}\tau_R$ is the Raman coefficient with $\tau_R$ the Raman shock time, $\phi$ is azimuthal angle in the cavity, $\Delta\omega_o \equiv \omega_{oD} - \omega_{oS}$, and $\Delta D \equiv D_{1D} - D_{1S}$. Also, $f_o \equiv \sqrt{\kappa_{\text{ext}}P_{\text{in}}}$ is the pump field amplitude where $\kappa_{\text{ext}}$ is the external coupling rate of the soliton mode and $P_{\text{in}}$ is the pump power. $G$ is the coupling strength between the soliton and dispersive wave mode families.

After considerable algebra, the spectral recoil $\Omega_{\text{Recoil}}$ induced by the dispersive wave can be obtained using moment analysis, as given by[26]

$$\Omega_{\text{Recoil}} = -\Lambda\tau_s|h_r|^2 \propto \frac{1}{(\Delta\omega')^2[(\Delta\omega_r - \Delta\omega')^2 + \frac{\kappa_D^2}{4}]}, \quad (19)$$

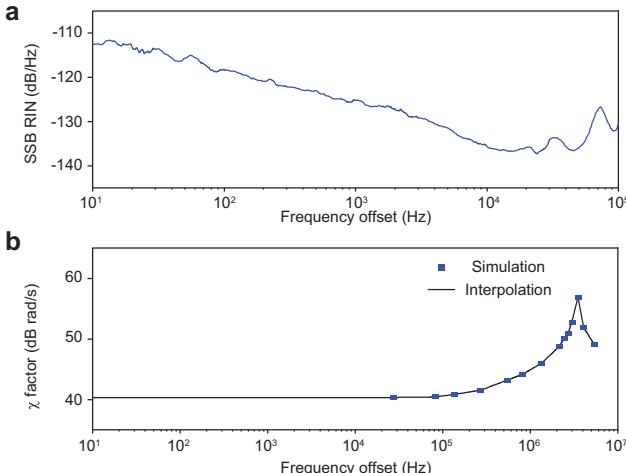

**Fig. 8 Influence of pump laser relative intensity noise (RIN). a** Measured SSB RIN of the pump laser. **b** Simulated noise transduction factor $\chi(f)$ between pump laser RIN and soliton repetition rate.

with $\Lambda$

$$\Lambda = \frac{\pi r \kappa_D g_S D_{1S}^2}{\kappa_S D_{2S}}. \tag{20}$$

where $r$ is the relative mode index of the mode in which the dispersive wave emits. Combined with the Raman-induced SSFS[16]

$$\Omega_{Raman} = -\frac{8\tau_R D_{2S}}{15\kappa_S D_{1S}^2 \tau_s^4}, \tag{21}$$

the overall soliton spectral shift yields

$$\Omega = \Omega_{Raman} + \Omega_{Recoil}. \tag{22}$$

In addition, the detuning $\delta\omega$ is obtained by[16,26]

$$\delta\omega = \frac{D_{2S}}{2D_{1S}^2}\left(\frac{1}{\tau_s^2} + \Omega^2\right). \tag{23}$$

It is noted $\Omega$ and $\tau_s$ are fitted through optical spectra of the soliton with a sech$^2$ envelope. All parameters that are required to describe the system are obtained either from direct measurement or by fitting experimental data. Critical parameters are: $\tau_R = 2.7$ fs, $g_S = 7.9 \times 10^{-4}$ rad/s. Fitted parameters include $G/2\pi = 12 \pm 2$ MHz and $\Delta\omega'/2\pi = 16 \pm 2$ MHz. Based on these parameters, the transduction factors are calculated numerically as shown in Fig. 5d. The analytical model is further verified with numerical simulation in the Supplementary note I.

**Impact of pump intensity noise.** The impact of the relative intensity noise (RIN) from the pump laser on the soliton repetition rate is obtained by numerical simulation of the above-mentioned coupled Lugiato–Lefever equations. Specifically, a sinusoidal perturbation at frequency $f$ is applied on the pump so that

$$f_o = f_s\left(1 + \frac{\varepsilon}{2}\sin 2\pi f t\right). \tag{24}$$

By tracking the motion of soliton peak position, the change of repetition rate can be revealed which further gives the following noise coupling coefficient

$$\chi(f) = |\frac{\partial \omega_{rep}}{\partial \epsilon}|. \tag{25}$$

Therefore, with the measured pump RIN (Fig. 8), the RIN-induced phase noise can be derived as shown in Fig. 3d.

**Quantum timing jitter.** Quantum timing jitter of soliton microcombs originates from vacuum fluctuations in each cavity mode. The predicted SSB phase noise spectral density of the detected soliton pulse stream at the repetition rate is given by[14]

$$S_Q(f) = \frac{\pi g_S}{6}\sqrt{\frac{2D_{2S}}{\delta\omega}}\left[\frac{\pi^2 \kappa_S}{16\delta\omega} + \left(1 + \frac{4\pi^2 f^2}{\kappa_S^2}\right)^{-1}\frac{\delta\omega}{\kappa_S}\right]. \tag{26}$$

## Data availability

The data that support the plots within this paper and other findings of this study are available on figshare (https://doi.org/10.6084/m9.figshare.13513995). All other data used in this study are available from the corresponding author upon reasonable request.

## Code availability

The codes that support the findings of this study are available from the corresponding authors upon reasonable request.

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

## Acknowledgements

We thank S. Papp at NIST for helpful comments during the preparation of this manuscript, N. M. Kondratiev for discussion on simulation of thermorefractive noise, as well as X. Zhang for help on simulation of the dispersive wave. The authors gratefully acknowledge the Defense Advanced Research Projects Agency under the APhI program (Award FA9453-19-C-0029) and the Kavli Nanoscience Institute. Q.-X.J. thanks the Caltech SURF program.

## Author contributions

Experiments were conceived and designed by Q-F.Y., Q-X.J. and K.V. Measurements are performed by Q-F.Y. and Q-X.J. with assistance from L.W., B.S., C.B. and Z.Y. Devices are fabricated and packaged by L.W. Modelling is performed by Q-F.Y., Q-X.J., H.W. and B.S. All authors participated in preparing the manuscript.

## Competing interests

The authors declare no competing interests.
