## [Peer Review File · Nature Communications]

Reviewers' Comments:

Reviewer #1:

Remarks to the Author:

In this manuscript, authors reported a miniature microwave source by means of the soliton microcomb technique. They carefully characterized the noise spectrum of the repetition frequency of the microcavity soliton, particularly around the so-called "quiet point" of the laser cavity detuning where the dispersive wave and Raman coupled noise can be counterbalanced. Given this situation, the phase noise of the soliton repetition rate is much reduced, to a record low level among chip scale soliton microcomb platforms, and the system reaches the limitation of thermorefractive noise (TRN). On the latter topic, authors further studied the coupling of TRN between a higher order mode of the cavity and the soliton mode family.

In my view, the results of the presenting work are solid, with fruitful details and supporting theories. Although the high-order mode coupled TRN seems not a dominant noise source compared with the fundamental one, especially at low offset frequencies (<1 kHz), this effect was carefully derived here for the first time.

However, my concern is that overall the novelty of the presenting work is not obvious. This work is the one after several others reporting on the soliton microcomb used for microwave signal generation. As such, the methodology including the quiet point operation regime in the presenting work has already been tested in previous works, particularly in the work by Lucas et al., "Ultralow-noise photonic microwave synthesis using a soliton microcomb-based transfer oscillator," *Nature Communications*, 11, 374, 2020 (In fact this work is highly related to the present one). In microcavities, the TRN limitation to the microwave signal is also well known, and clearly, with an ultrahigh Q factor, the TRN has limited impact such that the signal noise can be improved, as expected. Nevertheless, there remains a gap in the noise performance between the presenting work and a crystalline cavity microcomb (Ref. 4). Since the latter still holds the record of the lowest phase noise (ca. -110 dBc/Hz@1kHz) of soliton microcomb and the system was also miniature, the representing claim of a "record low" phase noise, though better than other chip based platforms, is degraded.

To conclude, there is no doubt the presenting manuscript is of high quality and is educational for people who are not familiar with the fundamentals of soliton microcombs. But given that a similar work has just been published ahead of the presenting one and the current results are not clearly outstanding, I may not suggest the work to be published in *Nature Communications*.

Reviewer #2:

Remarks to the Author:

The manuscript entitled 'Dispersive-wave induced noise limits in miniature soliton microwave sources' reports on a detailed investigation of the noise processes of a microcomb-based microwave source operating at a so-called quiet point. This quiet point occurs as a consequence of a delicate balance between Raman-induced soliton self-frequency shift, and spectral recoil associated with the emission of dispersive wave into a non-soliton transverse mode, as nicely demonstrated by the authors in ref. [20]. As a result, the detuning noise couples minimally to the soliton repetition rate and the microwave noise is significantly suppressed.

Using such a quiet point to reduce the noise of the resulting microwave signal is not new in itself (ref. [29] in the main text). However, I believe the novelty of the current manuscript lies in revealing and characterizing a new noise mechanism, which originates from correlations (or lack of) between the thermal fluctuations of the competing transverse modes. Such effect can arise whenever multiple transverse modes interact. This noise mechanism is thus likely to be of a broad interest to studies concerning micro- and nano-scale multimode systems, especially as the fundamental noise limit is approached.

In my opinion, the manuscript merits publication in Nature Communications. However, I would first like to invite the authors to consider my comments below:

- a. Since the work concerns photonic-chip-based microwave sources, it would be appropriate to introduce ref. [33-36] earlier to give audience a broader overview of progress that is in parallel to microcomb technology.
- b. To appeal to broad audience of Nature Communications, it would be pertinent to compare/contrast dispersive waves phase-matched by higher order dispersion (same mode family), with those arising from intermodal interaction. Ultrafast researchers more often associate dispersive waves with the former process in non-resonant single-pass waveguides. The statement 'These waves are emitted when solitons radiate into resonator modes that do not belong to the soliton-forming mode family' with no further elaboration may cause confusion.
- c. In addition to ref. [19,42,43], Jang et al. Opt. Lett. 39, 5503 (2014) reports on dispersive wave generation in a driven fiber resonator and deserves to be acknowledged in my opinion.
- d. In discussion, the authors claim the measured phase noise is a record low among photonic-chip-based microwave sources. While I do not doubt the claim, it would be good to compare the performance of their system with other systems, possibly in a tabular form.
- e. Please clarify how detuning is measured. I see in the methods that the authors use Eq (23) but there is no clear reference to that expression in the main text.
- f. In Fig. 3e, phase noise reductions initially exceed the expected black dotted curve at 500 Hz and 1 kHz. Why is that? Is that due to the correlation-induced reduction in TRN? Please also refer to the next point.
- g. On a related note, how does the conclusion from the study of intermode TRN relate to the results of Fig. 3d and e? Specifically, because the correlation R is positive at lower Fourier frequencies, the intermode TRN is reduced according to Eq (4) and is supported by Fig. 5b. However, noise suppression factor saturates more strongly at lower frequencies in Fig. 3d and e, and seem to contradict Eq (4).
- h. Please clarify what the authors mean by 'spectral center' of the comb. Is it the peak of the sech² envelope fit, or is it the first-order moment of the measured spectrum?
- i. Just a remark. Figure 4b is very nice and illustrative!

Other minor comments:

1. Just below Eq (1), strictly D1 is $2\pi \text{FSR}$.
2. Page 2 right column, there is repeated 'of the of the'.
3. In main text and caption of Fig. 3, the authors refer to dashed orange curve in Fig. 3d. It looks like a dashed red curve.
4. In section 'Thermal noise in the dispersive wave', the first sentence 'Constant heat exchange associated with thermal equilibrium' reads contradictory because at thermal equilibrium, there should be no heat exchange. Perhaps more appropriate to clarify e.g. 'fluctuations about thermal equilibrium' or equivalent.
5. What does wide tilde in Eq (11) and (16) signify?

Reviewer #3:

Remarks to the Author:

Please see the attached file for reviewer comments.

Id: ncomms-20-39020
Title: Dispersive-wave induced noise limits in miniature soliton microwave sources
Authors: Qi-Fan Yang *et al.*

In this manuscript the authors use a frequency comb generated by continuous wave pumping of an ultra-high quality microresonator to generate a microwave frequency of about 15 GHz. In particular, the authors investigate and present phase fluctuations of the microwave beat frequency when the system is operated around the so-called quiet point, where the Raman-induced self-frequency shift of the soliton is balanced by the dispersive-wave induced recoil. Both are a function of the frequency detuning defined as the difference between the frequency of the cavity mode nearest to the pump frequency and the pump frequency.

A calibration tone, generated as a side band of the pump frequency, is used to compare the phase noise measured for different detuning frequencies. The calibration tone is used to determine the expected noise suppression for different detuning frequencies. It is found in the experiment that the actual noise suppression is less than that derived from the calibration tone at and around the quiet point. From this the authors conclude that other noise sources are limiting the phase noise around the quiet point.

The authors subsequently consider various noise sources, like pump intensity noise, noise induced by quantum vacuum fluctuations and temperature-induced fluctuation in the free-spectral range of the resonator and conclude that these are all too small to explain the measured phase noise of the microwave beat frequency. The authors conclude, by comparing theoretical and simulation results with measurements, that dispersive wave induced noise originating from fundamental intermode thermorefractive noise is the limiting noise source for their device.

Overall, the manuscript is well structured and written. The research is novel, of high interest, and the claims are well formulated and supported by appropriate measurements and simulations. However, the authors provide uncertainty in measured/calculated values only for the Allan deviation (fig. 2c), while such information is absent for all other results presented. It would benefit the reader to have more insight in the accuracy of the measured and calculated values presented in this manuscript.

Other minor issues that need to be clarified are:

1. ω_0 is defined as the frequency of the cavity mode pumped by the laser. Is this the cold-cavity frequency or the hot-cavity frequency of the resonance?
2. Definition of $\Delta\omega$ is not clear. Is this parameter the difference between hybridized modes or non-hybridized modes? The main text suggests this to be the difference between non-hybridized modes, while the supplementary information uses inline formula " $\omega'_{r,D} = \omega_{r,S} - \Delta\omega$ ", suggesting that $\Delta\omega$ is the frequency difference between a hybridized and non-hybridized mode.
3. Formula S1: From the reference given for formula S1 and formula S1 itself, I would assume $\Delta\omega$ to be the frequency difference of the non-hybridized modes. However, if true, formula S1 assumes no loss or at least no loss difference between the non-

hybridized resonances, although this is an ultra-high-Q resonator and losses are extremely low, is the difference still considered significantly small compared to the coupling term G and/or $\Delta\omega$ for equation S1 to hold?

4. "As a benchmark of the stability ..." The authors provide an Allan deviation analysis of the soliton repetition rate and retrieve the minimum deviation. However, this value is not placed in a context. How does this device compare to other comb-based microwave sources, in particular how does it compare to the electronic microwave source used for the locking demonstration?
5. The calibration tone is visible as a marker in figure 3d, but not in the measured traces of the SSB phase noise. Do these represent two different measurements, i.e., one with calibration tone and a subsequent measurement without calibration tone?
6. "Their Q factors are also measured, as shown in Fig. 4d". Why is this figure included? The Q factors seem not to be used elsewhere in the manuscript.
7. Equation 4 requires a reference to the methods section.
8. Authors mention that the microwave beat signal could track the external microwave source over a range of 30 kHz. The limits of such tracking are not discussed. What limits this tuning range? What is the minimum tuning step that this system can realize?
9. "The laser scan is precisely measured by a radio-frequency calibrated Mach-Zehnder interferometer". The reference used provides a similar statement without providing more details. The reference should point to a paper explaining this measurement technique. The current reference is inappropriate.
10. "averaged temperature of an optical mode". I would not call this a temperature of an optical mode but an "optical mode weighted average temperature".
11. It is not completely clear what the density $q(r)$ represents. Does $E(r)$ in eq. 9 represent the transverse distribution of the electric field of a single transverse mode or the total transverse distribution if multiple transverse modes are present? Also, as we are dealing with a relatively broad frequency comb, even for a single transverse mode, the mode profile will slightly vary for different resonant frequencies. Is this dependency included or neglected?
12. It is unclear what the tilde in eq. 11 represents. If it has the same meaning as in eq 16, it should represent the Fourier transform, however that is only introduced with eq. 16.
13. Comparing eq. 11 with the references shows a discrepancy by a factor of 2. Is this a typo?
14. In equation 12, the f in $\frac{\hbar W_{diss}}{\pi f}$ is actually the amplitude squared of the entropy injection F_0^2 according to reference 17. Also, again there seem to be a factor of 2 discrepancy with the references.
15. In equation 14 the Fourier transform is indicated by \mathcal{F} while in eq. 16 (and eq. 11) a tilde above the variable is used to indicate the Fourier transform. Two different notations are used, where only one should be needed.
16. The density $q(r)$ seems to be missing in the source term of equation 16. The source term should be the Fourier transform of eq. 10 multiplied by $i\omega T_0$ conform equation 7.
17. Equations 17 & 18 only contains a linear coupling term. Nonlinear coupling (e.g., cross-phase modulation) can also be present. Why has this not been included in the model,

i.e., why is this effect considered weak enough to neglect even for the dispersive wave (eq. 18)?

18. It is stated that the detuning noise is correlated to the noise in the error signal of the PDH, which makes sense. However, it is unclear how the conversion from PDH-error signal to detuning noise is performed. The text only mentions that it is “extracted from the residual error signal in the locking loop”

Small typos:

“(see Fig. 1.” Missing closing bracket

“being pump by optical”. Should be “being pumped by optical”

“of the of the”. Repetition

“maintext” Should be “main text”.

Supplemental information

“filed”. Should be “field”. Furthermore, Fig. s1a indicates that the power is plotted and not the field and it is not explained what “cavity angle” means (horizontal label in Fig. s1a).

Response to reviewer #1:

In this manuscript, authors reported a miniature microwave source by means of the soliton microcomb technique. They carefully characterized the noise spectrum of the repetition frequency of the microcavity soliton, particularly around the so-called “quiet point” of the laser cavity detuning where the dispersive wave and Raman coupled noise can be counterbalanced. Given this situation, the phase noise of the soliton repetition rate is much reduced, to a record low level among chip scale soliton microcomb platforms, and the system reaches the limitation of thermorefractive noise (TRN). On the latter topic, authors further studied the coupling of TRN between a higher order mode of the cavity and the soliton mode family.

In my view, the results of the presenting work are solid, with fruitful details and supporting theories. Although the high-order mode coupled TRN seems not a dominant noise source compared with the fundamental one, especially at low offset frequencies (<1 kHz), this effect was carefully derived here for the first time.

However, my concern is that overall the novelty of the presenting work is not obvious. This work is the one after several others reporting on the soliton microcomb used for microwave signal generation. As such, the methodology including the quiet point operation regime in the presenting work has already been tested in previous works, particularly in the work by Lucas et al., “Ultralow-noise photonic microwave synthesis using a soliton microcomb-based transfer oscillator,” Nature Communications, 11, 374, 2020 (In fact this work is highly related to the present one). In microcavities, the TRN limitation to the microwave signal is also well known, and clearly, with an ultrahigh Q factor, the TRN has limited impact such that the signal noise can be improved, as expected. Nevertheless, there remains a gap in the noise performance between the presenting work and a crystalline cavity microcomb (Ref. 4). Since the latter still holds the record of the lowest phase noise (ca. -110 dBc/Hz@1kHz) of soliton microcomb and the system was also miniature, the representing claim of a “record low” phase noise, though better than other chip-based platforms, is degraded.

To conclude, there is no doubt the presenting manuscript is of high quality and is educational for people who are not familiar with the fundamentals of soliton microcombs. But given that a similar work has just been published ahead of the presenting one and the current results are not clearly outstanding, I may not suggest the work to be published in Nature Communications.

Reply: We thank the reviewer for his/her appreciation of our work. We agree that the soliton microcombs generated in crystalline resonators feature outstanding performance due to their ultrahigh-Q factors and low thermo-optic coefficients. However, it remains a challenge to transfer these devices to photonic chips. The silica resonator used in our work holds the record Q factor among all chip-based platforms, and its integration with a photonic waveguide has also been demonstrated [K. Yang et al, Bridging ultrahigh-Q devices and photonic circuits, Nature Photonics 12, 297 (2018)]. Indeed, our results surpass all other chip-based soliton microcombs and optoelectronic oscillators. To highlight this point, in the revised manuscript a new table (also attached here) is included to compare the phase noise and Allan deviation of several state-of-the-art optoelectronic oscillators.

Besides the record chip-based performance, a new noise source of soliton microcombs is revealed in this work. While the dispersive wave induced ‘quiet point’ mechanism has been used to reduce the impact of pump phase noise [Nature Communications, 8(1), 14869; Nature Communications, 11(1); Nature Photonics, 14(8), 486–491; Physical Review Letters,

125(15), 153901], limits of this technique have not yet been explored. In this work we have shown that while quiet point reduction can exceed 36 dB, intermode thermorefractive noise channels through the dispersive wave involved with the quiet point operation, itself. And, indeed this limits phase noise suppression by the quiet point. Since researchers have been striving to generate low-noise soliton microcombs to facilitate a wide range of applications, understanding this noise mechanisms is important. Hence, we believe the key findings of our work are novel and are of great interest to the community of optical frequency combs and microwave photonics.

Table 1 Miniature photonic-based microwave oscillators					
Material	Configuration	Phase noise			Ref #
		Carrier freq. (GHz)	SSB phase noise (dBc/Hz, scaled to 15.2 GHz)		
			1 kHz	100 kHz	
SiO ₂ (this work)	Bright soliton	15.2	-90	-140	
MgF ₂	Bright soliton	14.1	-121	-155	35
Si ₃ N ₄	Bright soliton	19.6	-82	-132	7
Si ₃ N ₄	Dark soliton	5.4	-76	-131	8
SiO ₂	SBS	21.7	-71	-113	10
Si ₃ N ₄	SBS	21.8	-55	-102	13
chalcogenide-on-silicon	SBS	40.0	-93	-110	39
Relative Allan deviation					
Material	Configuration	Min. Adev. gate time (ms)	Min. Adev.	Adev. @ 1 s	Ref #
SiO ₂ (this work)	Bright soliton	50	6×10 ⁻¹¹	2×10 ⁻¹⁰	
MgF ₂	Bright soliton	200	5×10 ⁻¹²	10 ⁻¹¹	4
Si ₃ N ₄	Bright soliton	1	3×10 ⁻⁹	8×10 ⁻⁸	7
SiO ₂	SBS	20	10 ⁻⁹	10 ⁻⁸	10

Response to reviewer #2:

The manuscript entitled 'Dispersive-wave induced noise limits in miniature soliton microwave sources' reports on a detailed investigation of the noise processes of a microcomb-based microwave source operating at a so-called quiet point. This quiet point occurs as a consequence of a delicate balance between Raman-induced soliton self-frequency shift, and spectral recoil associated with the emission of dispersive wave into a non-soliton transverse mode, as nicely demonstrated by the authors in ref. [20]. As a result, the detuning noise couples minimally to the soliton repetition rate and the microwave noise is significantly suppressed.

Using such a quiet point to reduce the noise of the resulting microwave signal is not new in itself (ref. [29] in the main text). However, I believe the novelty of the current manuscript lies in revealing and characterizing a new noise mechanism, which originates from correlations (or lack of) between the thermal fluctuations of the competing transverse modes. Such effect can arise whenever multiple transverse modes interact. This noise mechanism is thus likely to be of a broad interest to studies concerning micro- and nano-scale multimode systems, especially as the fundamental noise limit is approached.

In my opinion, the manuscript merits publication in Nature Communications. However, I would first like to invite the authors to consider my comments below:

Reply: We thank the reviewer for his/her recommendation for publication.

a. Since the work concerns photonic-chip-based microwave sources, it would be appropriate to introduce ref. [33-36] earlier to give audience a broader overview of progress that is in parallel to microcomb technology.

Reply: We thank the reviewer for this suggestion and have included these references in the introduction in the revised manuscript.

b. To appeal to broad audience of Nature Communications, it would be pertinent to compare/contrast dispersive waves phase-matched by higher order dispersion (same mode family), with those arising from intermodal interaction. Ultrafast researchers more often associate dispersive waves with the former process in non-resonant single-pass waveguides. The statement ‘These waves are emitted when solitons radiate into resonator modes that do not belong to the soliton-forming mode family’ with no further elaboration may cause confusion.

Reply: we thank the reviewer for this suggestion. We have added the following sentence to clarify mechanism that can produce dispersive waves: “Dispersive waves can emerge as a result of higher-order dispersion [23,24], supermodes [25], or when solitons radiate into resonator modes that do not belong to the soliton-forming mode family.” in the revised manuscript. We also have noted in the discussion a way to reduce the noise through use of dispersive waves formed in the same mode longitudinal mode family: “First, use of dispersive waves within the same longitudinal mode family (as formed by higher order dispersion^{23,24,42,43}) could be investigated. In this case, better overlap of the dispersive wave modal profile with the soliton mode would be expected to reduce the dispersive wave noise.”

c. In addition to ref. [19,42,43], Jang et al. Opt. Lett. 39, 5503 (2014) reports on dispersive wave generation in a driven fiber resonator and deserves to be acknowledged in my opinion.

Reply: We thank the reviewer for the comment. We have included this citation in the revised manuscript.

d. In discussion, the authors claim the measured phase noise is a record low among photonic-chip-based microwave sources. While I do not doubt the claim, it would be good to compare the performance of their system with other systems, possibly in a tabular form.

Reply: We thank the reviewer for this suggestion. We have included a table in the revised manuscript (also included here). The phase noise values are all scaled to 15.2 GHz. Our work does show best noise performance among all chip-based platforms.

Table 1 Miniature photonic-based microwave oscillators					
Phase noise					
Material	Configuration	Carrier freq. (GHz)	SSB phase noise (dBc/Hz, scaled to 15.2 GHz)		Ref #
			1 kHz	100 kHz	
SiO ₂ (this work)	Bright soliton	15.2	-90	-140	
MgF ₂	Bright soliton	14.1	-121	-155	35
Si ₃ N ₄	Bright soliton	19.6	-82	-132	7
Si ₃ N ₄	Dark soliton	5.4	-76	-131	8
SiO ₂	SBS	21.7	-71	-113	10
Si ₃ N ₄	SBS	21.8	-55	-102	13
chalcogenide-on-silicon	SBS	40.0	-93	-110	39
Relative Allan deviation					
Material	Configuration	Min. Adev. gate time (ms)	Min. Adev.	Adev. @ 1 s	Ref #
SiO ₂ (this work)	Bright soliton	50	6×10 ⁻¹¹	2×10 ⁻¹⁰	
MgF ₂	Bright soliton	200	5×10 ⁻¹²	10 ⁻¹¹	4
Si ₃ N ₄	Bright soliton	1	3×10 ⁻⁹	8×10 ⁻⁸	7
SiO ₂	SBS	20	10 ⁻⁹	10 ⁻⁸	10

e. Please clarify how detuning is measured. I see in the methods that the authors use Eq (23) but there is no clear reference to that expression in the main text.

Reply: The detuning is calculated based on the fitted spectral center and pulse width of the soliton with Eq (23). We have added a sentence “Here the detuning $\delta\omega$ is calculated based on equation (23) in Methods” in the revised manuscript.

f. In Fig. 3e, phase noise reduction initially exceed the expected black dotted curve at 500 Hz and 1 kHz. Why is that? Is that due to the correlation-induced reduction in TRN? Please also refer to the next point.

Reply: we thank the reviewer for this helpful comment. We believe that this is an artifact of using a calibration tone at an offset frequency of 10 kHz and then making noise measurements at a series of other offset frequencies (500 Hz, 1 kHz and 8 kHz). Specifically, the artifact does not appear in the 8 kHz data. In the revised manuscript, we have added a comment as follows: “As an aside, the quiet-point-induced phase noise reduction is also slightly higher than indicated by the calibration tone for lower noise suppression levels (when measured at 500 Hz and 1 kHz offsets). This could result from possible instrument calibration error associated with calibration using a 10 kHz tone.”.

g. On a related note, how does the conclusion from the study of intermode TRN relate to the results of Fig. 3d and e? Specifically, because the correlation R is positive at lower Fourier frequencies, the intermode TRN is reduced according to Eq (4) and is supported by Fig. 5b. However, noise suppression factor saturates more strongly at lower frequencies in Fig. 3d and e, and seem to contradict Eq (4).

Reply: Even though there is stronger correlation in the intermode TRN at lower offset frequencies, the frequency response of the TRN noise noise strongly increases as offset frequency decreases. And this increase is so strong that it dominates the improved intermode TRN correlation discussed later. This is indeed a confusing point and we have added the following discussion in the main text to clarify.

“Notice that despite the improved correlation of the intermode TRN at lower offset frequencies, it still dominates the microwave phase noise measured in Fig. 3d,e. This happens because the TRN noise rises very rapidly as offset frequency decreases, even overcoming the improving correlation of TRN between the dispersive wave mode and soliton forming mode.”

h. Please clarify what the authors mean by ‘spectral center’ of the comb. Is it the peak of the sech2 envelope fit, or is it the first-order moment of the measured spectrum?

Reply: ‘Spectral center’ is the peak of the sech2 envelope fit. We have rephrased the first appearance of this expression and defined it: “.....center frequency of the soliton spectrum (which has an overall sech2 envelope)..”

i. Just a remark. Figure 4b is very nice and illustrative!

Reply: We thank the reviewer for the remark.

Other minor comments:

Just below Eq (1), strictly $D1$ is $2\pi FSR$.

Reply: we have corrected this definition.

Page 2 right column, there is repeated ‘of the of the’.

Reply: we have corrected the typo.

In main text and caption of Fig. 3, the authors refer to dashed orange curve in Fig. 3d. It looks like a dashed red curve.

Reply: We have corrected the typo to “dashed red curve”.

In section ‘Thermal noise in the dispersive wave’, the first sentence ‘Constant heat exchange associated with thermal equilibrium’ reads contradictory because at thermal equilibrium, there should be no heat exchange. Perhaps more appropriate to clarify e.g. ‘fluctuations about thermal equilibrium’ or equivalent.

Reply: We have changed the expression to “fluctuations associated with thermal equilibrium”.

What does wide tilde in Eq (11) and (16) signify?

Reply: The tildes signify Fourier transform of the quantity. In the revised manuscript, we have unified the notations of Fourier transform to be scripted \mathcal{F} .

Response to reviewer #3:

In this manuscript the authors use a frequency comb generated by continuous wave pumping of an ultra-high quality microresonator to generate a microwave frequency of about 15 GHz. In particular, the authors investigate and present phase fluctuations of the microwave beat frequency when the system is operated around the so-called quiet point, where the Raman-induced self-frequency shift of the soliton is balanced by the dispersive-wave induced recoil. Both are a function of the frequency detuning defined as the difference between the frequency of the cavity mode nearest to the pump frequency and the pump frequency.

A calibration tone, generated as a side band of the pump frequency, is used to compare the phase noise measured for different detuning frequencies. The calibration tone is used to determine the expected noise suppression for different detuning frequencies. It is found in the experiment that the actual noise suppression is less than that derived from the calibration tone at and around the quiet point. From this the authors conclude that other noise sources are limiting the phase noise around the quiet point.

The authors subsequently consider various noise sources, like pump intensity noise, noise induced by quantum vacuum fluctuations and temperature-induced fluctuation in the free spectral range of the resonator and conclude that these are all too small to explain the measured phase noise of the microwave beat frequency. The authors conclude, by comparing theoretical and simulation results with measurements, that dispersive wave induced noise originating from fundamental intermode thermorefractive noise is the limiting noise source for their device.

Overall, the manuscript is well structured and written. The research is novel, of high interest, and the claims are well formulated and supported by appropriate measurements and simulations.

However, the authors provide uncertainty in measured/calculated values only for the Allan deviation (fig. 2c), while such information absents for all other results presented. It would benefit the reader to have more insight in the accuracy of the measured and calculated values presented in this manuscript.

Reply: We thank the reviewer for his/her appreciations of our work, as well as the expert comments. We have included a detailed analysis of uncertainty in Methods of the revised manuscript. Error bars are also provided in the revised Fig3 and Fig 5.

Minor issues that need to be clarified are:

1. ω_0 is defined as the frequency of the cavity mode pumped by the laser. Is this the cold-cavity frequency or the hot-cavity frequency of the resonance?

Reply: ω_0 denotes cold-cavity frequency. We have clarified this point in the revised manuscript as “cold-cavity mode”.

2. Definition of $\Delta\omega$ is not clear. Is this parameter the difference between hybridized modes or non-hybridized modes? The main text suggests this to be the difference between non-hybridized modes, while the supplementary information uses inline formula “ $\omega'_{r,D} = \omega_{r,S} - \Delta\omega$ ” suggesting that $\Delta\omega$ is the frequency difference between a hybridized and non-hybridized mode.

Reply: we thank the reviewer for raising this question. In the previous version of the manuscript, we were assuming that the mode hybridization is weak, and we were not distinguishing their differences. We have added discussion in the revised manuscript, and it is shown that the $\Delta\omega$ in the previous version should be frequency difference between a hybridized and non-hybridized mode. To clarify this point, in the revised manuscript, we use $\Delta\omega$ to represent difference between the non-hybridized modes and $\Delta\omega'$ to represent the frequency difference between the hybridized dispersive wave mode and non-hybridized soliton mode.

3. Formula S1: From the reference given for formula S1 and formula S1 itself, I would assume $\Delta\omega$ to be the frequency difference of the non-hybridized modes. However, if true, formula S1 assumes no loss or at least no loss difference between the non-hybridized

resonances, although this is an ultra-high-Q resonator and losses are extremely low, is the difference still considered significantly small compared to the coupling term G and/or $\Delta\omega$ for equation S1 to hold?

Reply: Yes, we are assuming that the difference is small compared to the couplings and detuning. A complete expression yields

$$\omega'_{rD} = \frac{\omega_{rS} + \omega_{rD}}{2} - \text{Real}\left[\sqrt{G^2 + \frac{4\Delta\omega^2 - (\kappa_S - \kappa_D)^2 + 4j(\kappa_S - \kappa_D)\Delta\omega}{16}}\right] = \frac{\omega_{rS} + \omega_{rD}}{2} - \sqrt{G^2 + \frac{\Delta\omega^2}{4}} \left[1 + O\left(\frac{(\kappa_S - \kappa_D)^2}{G^2 + \frac{\Delta\omega^2}{4}}\right)\right].$$

In over case, $(\kappa_D - \kappa_S)/2\pi = 2.17$ MHz, while the denominator $\sqrt{G^2 + \frac{\Delta\omega^2}{4}}/2\pi = 12.2$ MHz.

A comment is added in the Supplement to clarify this point.

4. “As a benchmark of the stability ...” The authors provide an Allan deviation analysis of the soliton repetition rate and retrieve the minimum deviation. However, this value is not placed in a context. How does this device compare to other comb-based microwave sources, in particular how does it compare to the electronic microwave source used for the locking demonstration?

Reply: We thank the reviewer for raising this question. We have compared both phase noise and Allan deviation of our result with those of several other platforms: crystalline microcomb, SiN microcomb, photonic Brillouin microwave synthesizer and an integrated opto-electrical oscillator. However, we could not quantify Allan deviation of the table-top electronic microwave source, as it is the most stable microwave reference in our lab.

Table 1 Miniature photonic-based microwave oscillators					
Material	Configuration	Phase noise			Ref #
		Carrier freq. (GHz)	SSB phase noise (dBc/Hz, scaled to 15.2 GHz)		
			1 kHz	100 kHz	
SiO ₂ (this work)	Bright soliton	15.2	-90	-140	
MgF ₂	Bright soliton	14.1	-121	-155	35
Si ₃ N ₄	Bright soliton	19.6	-82	-132	7
Si ₃ N ₄	Dark soliton	5.4	-76	-131	8
SiO ₂	SBS	21.7	-71	-113	10
Si ₃ N ₄	SBS	21.8	-55	-102	13
chalcogenide-on-silicon	SBS	40.0	-93	-110	39
Relative Allan deviation					
Material	Configuration	Min. Adev. gate time (ms)	Min. Adev.	Adev. @ 1 s	Ref #
SiO ₂ (this work)	Bright soliton	50	6×10 ⁻¹¹	2×10 ⁻¹⁰	
MgF ₂	Bright soliton	200	5×10 ⁻¹²	10 ⁻¹¹	4
Si ₃ N ₄	Bright soliton	1	3×10 ⁻⁹	8×10 ⁻⁸	7
SiO ₂	SBS	20	10 ⁻⁹	10 ⁻⁸	10

As shown in the figure, our result exceeds the performance of SiN microcomb and Brillouin microwave synthesizer, but is not as good as the Crystalline microcomb.

5. The calibration tone is visible as a marker in figure 3d, but not in the measured traces of the SSB phase noise. Do these represent two different measurements, i.e., one with calibration tone and a subsequent measurement without calibration tone?

Reply: They are the same measurements. We were using a signal source analyzer (R&S®FSUP Signal Source Analyzer) with spur rejection function. The spurs were simultaneously detected and do not appear in the plotted phase noise traces.

6. *“Their Q factors are also measured, as shown in Fig. 4d”. Why is this figure included? The Q factors seem not to be used elsewhere in the manuscript.*

Reply: The Q factors are measured for evaluating the loss of the modes κ_S and κ_D , which is further used for evaluating the noise transduction factor.

7. *Equation 4 requires a reference to the methods section.*

Reply: Thanks for the suggestion. We have added the reference in the revised version.

8. *Authors mention that the microwave beat signal could track the external microwave source over a range of 30 kHz. The limits of such tracking are not discussed. What limits this tuning range? What is the minimum tuning step that this system can realize?*

Reply: We thank the reviewer for the comment. It is limited by the pump power of the soliton, as the possible detuning range between the pump laser and cavity mode depends primarily on the pump power [Nature Physics, 13(1), 94–102]. Continuous tuning is feasible. We have included a sentence “This range is likely determined by the soliton existence range, which is, in turn, determined by the pump laser power 33.” in the revised manuscript.

9. *“The laser scan is precisely measured by a radio-frequency calibrated Mach-Zehnder interferometer”. The reference used provides a similar statement without providing more details. The reference should point to a paper explaining this measurement technique. The current reference is inappropriate.*

Reply: We thank the reviewer for noticing this. A more detailed description is included in [Optics Express, 20(24), 26337 (2012)]. We have added this citation.

10. *“averaged temperature of an optical mode”. I would not call this a temperature of an optical mode but an “optical mode weighted average temperature”.*

Reply: We thank the reviewer for suggesting a more accurate description. We have replaced the phrase “averaged temperature of an optical mode” with “optical mode weighted average temperature”.

11. *It is not completely clear what the density $q(r)$ represents. Does $q(r)$ eq. 9 represent the transverse distribution of the electric field of a single transverse mode or the total transverse distribution if multiple transverse modes are present?*

Reply: $q(r)$ represents the transverse distribution of the electric field of a single transverse mode.

Also, as we are dealing with a relatively broad frequency comb, even for a single transverse mode, the mode profile will slightly vary for different resonant frequencies. Is this dependency included or neglected?

Reply: We are neglecting this effect. We have run FEM simulations to verify this approximation over the measured wavelength range (1530-1570 nm).

We have evaluated the intermode TRN between the TM₀ modes at frequencies 191 THz and 195 THz. Both material and geometry dispersion are considered. This noise is considered to directly modulate the FSR of the resonator and impose a noise limit on its repetition rate. The results are plotted below, along with the measured phase noise.

12. It is unclear what the tilde in eq. 11 represents. If it has the same meaning as in eq 16, it should represent the Fourier transform, however that is only introduced with eq. 16.

Reply: we thank the reviewer for noticing this. This is a typo, and there should not be the tilde.

13. Comparing eq. 11 with the references shows a discrepancy by a factor of 2. Is this a typo?

Reply: We thank the reviewer for noticing this. It is not a typo. We were using $|\delta T|$ to represent amplitude of δT , which is different from the reference [Physics Letters A, 372(12), 1941–1944 (2008)]. In the revised manuscript, to unify the notation where the reference, we have changing the expression to $W_{diss} = \int \frac{k}{T_0} \langle (\Delta\delta T)^2 \rangle d^3r$, where $\langle \rangle$ denotes time averaging.

14. In equation 12, the f in $\frac{hrW_{diss}}{\pi f}$ is actually the amplitude squared of the entropy injection F_0^2 according to reference 17. Also, again there seems to be a factor of 2 discrepancy with the references.

Reply: Here, f is the frequency of the noise. F_0^2 is normalized as 1 in the calculations, and we have included F_0^2 in the denominator of equation (12) for classification in the revised manuscript. Here, we used single side band definition of power spectral density, while the references are using one-sided (two side band) definition. The “2” factor comes from the different definition.

15. In equation 14 the Fourier transform is indicated by F while in eq. 16 (and eq. 11) a tilde above the variable is used to indicate the Fourier transform. Two different notations are used, where only one should be needed.

Reply: We have unified the notations in the revised manuscript as fancy \mathcal{F} .

16. The density $q(r)$ seems to be missing in the source term of equation 16. The source term should be the Fourier transform of eq. 10 multiplied by $i\omega T_0$ from equation 7.

Reply: We thank the reviewer for pointing out this typo. We have corrected this.

17. Equations 17 & 18 only contain a linear coupling term. Nonlinear coupling (e.g., cross phase modulation) can also be present. Why has this not been included in the model, i.e., why is this effect considered weak enough to neglect even for the dispersive wave (eq. 18)?

Reply: We thank the reviewer for the comment. While the power of the dispersive wave could exceed that of most comb lines, its power is still negligible compared with the peak power of the soliton. That is, its impact to the soliton through cross phase modulation is minor compared to the self phase modulation induced by the soliton itself. We have evaluated the cross-phase modulation contribution here.

Frequency shifted by cross phase modulation for the dispersive wave mode is evaluated by $g_c |E_S|^2$, where the cross-phase modulation factor, $g_c = \frac{r\omega_0 n_2 D_1}{\pi n_0 A_c}$, where the nonlinear effective

$$\text{area } A_c = \frac{\int |E_S|^2 d^2r \int |E_D|^2 d^2r}{\int |E_S|^2 |E_D|^2 d^2r}.$$

FEM simulation gives $A_c = 186 \mu\text{m}^2$, thus $g_c = 3.87 \times 10^{-4}$ rad/s. Based on numerical simulations, the cross-phase modulation on dispersive wave is less than 0.5 MHz during the measurement range, which is 1/6 of the dispersive mode linewidth (~3 MHz).

18. It is stated that the detuning noise is correlated to the noise in the error signal of the PDH, which makes sense. However, it is unclear how the conversion from PDH-error signal to detuning noise is performed. The text only mentions that it is “extracted from the residual error signal in the locking loop”.

Reply: We thank the reviewer for the comment. We were using the following steps:

1. The conversion relation between the error in and error output port of the servo box (Vescent photonics D2-125) is characterized before the locking turns on;
2. When the soliton is locked, signal of the error output port is measured with a signal analyzer, while locking parameters (e.g. gain and bandwidth) remain the same;
3. The correspondence between error in voltage and frequency scanned by the laser piezo is evaluated by turning the reference voltage of the servo with a function generator.

Small typos:

“(see Fig. 1.” Missing closing bracket

“being pump by optical”. Should be “being pumped by optical”

“of the of the”. Repetition

“maintext” Should be “main text”.

Supplemental information

“fild”. Should be “field”. Furthermore, Fig. s1a indicates that the power is plotted and not the field and it is not explained what “cavity angle” means (horizontal label in Fig. s1a).

Reply: We have corrected these typos.

Thank you again for your detailed review and comments.

Reviewers' Comments:

Reviewer #1:

Remarks to the Author:

I have read through the reply by authors and their revised manuscript as well as the suppl. file. As I have marked in the first review, this work consists of solid theoretical analysis and key experiments to verify authors finding of an intermode thermorefractive noise (TRN) regime. Also, authors now have included a table of comparison regarding the phase noise and the Allen deviation among several other platforms for photonic microwave generation, and their current results represent the best performance in photonic chip scale devices.

I am satisfied with authors reply and I appreciate their work in continuously improving the manuscript during the review process. Although I may retain my concern on the broad interest of this work, I have no objection to the acceptance of current manuscript by Nature Communications.

Reviewer #2:

Remarks to the Author:

It was a pleasure to read the revised manuscript. I only have several minor additional points

1. The authors claim to have measured phase noise using PDH locking of the pump laser at the top of page 7, and that the measurement is consistent with their expectation. It would be helpful to present the measurement data, if available.
2. In Eq. (5), S_Q and S_P are not defined until much later in the Methods. Though clear from the context, please define them explicitly in the text.
3. In Eq. (19), please use the notations which are consistent with Eq. (2). In particular, I believe the authors are making $\kappa_B \sim \kappa_{r-}$.
4. Please define ' r ' in Eq. (20).

I am happy to recommend the publication of the revised manuscript in Nature Communications.

Reviewer #3:

Remarks to the Author:

After having read the revised manuscript and authors rebuttal letter, I find that the authors have addressed all of the reviewers comments except for the following minor issue.

The authors specify (below eq. 23) $G/2\omega\pi = 11.05 \pm 2.03$ MHz, which is not an appropriate use of number of significant digits. This should be 11 ± 2 MHz. Likewise, for other values of other parameters.

With this corrected, I find the manuscript suitable for publication.

Response to reviewer1:

I have read through the reply by authors and their revised manuscript as well as the suppl. file. As I have marked in the first review, this work consists of solid theoretical analysis and key experiments to verify authors finding of an intermode thermorefractive noise (TRN) regime. Also, authors now have included a table of comparison regarding the phase noise and the Allen deviation among several other platforms for photonic microwave generation, and their current results represent the best performance in photonic chip scale devices.

I am satisfied with authors reply and I appreciate their work in continuously improving the manuscript during the review process. Although I may retain my concern on the broad interest of this work, I have no objection to the acceptance of current manuscript by Nature Communications.

Reply: We thank the reviewer for his/her appreciation for our efforts, as well as recommendation for publication.

Response to reviewer2:

It was a pleasure to read the revised manuscript. I only have several minor additional points. 1. The authors claim to have measured phase noise using PDH locking of the pump laser at the top of page 7, and that the measurement is consistent with their expectation. It would be helpful to present the measurement data, if available.

Reply: We have included the result using PDH method in the Supplementary Information. We have included a phrase "As expected the measured noise spectrum showed a limitation consistent with the dispersive wave noise (see Methods)." in the revised manuscript.

2. In Eq. (5), S_Q and S_P are not defined until much later in the Methods. Though clear from the context, please define them explicitly in the text.

Reply: We thank the reviewer for the suggestion. We have included a phrase "where S_Q is the quantum noise limit, and S_P is noise transferred from intensity noise of the pump laser" in the revised manuscript.

3. In Eq. (19), please use the notations which are consistent with Eq. (2). In particular, I believe the authors are making $\kappa_B \sim \kappa_{\{r\}}$.

Reply: We thank the reviewer for the suggestion. The notations have been unified as κ_B in the revised manuscript.

4. Please define 'r' in Eq. (20).

Reply: We thank the reviewer for the suggestion. We have included a phrase "where r is the relative mode index of the mode in which the dispersive wave emits" in the revised manuscript.

I am happy to recommend the publication of the revised manuscript in Nature Communications.

Reply: We thank the reviewer for his/her recommendation for publication.

Response to reviewer3:

After having read the revised manuscript and authors rebuttal letter, I find that the authors have addressed all of the reviewers comments except for the following minor issue. The authors specify (below eq. 23) $G/twopi = 11.05 \pm 2.03$ MHz, which is not an appropriate use of number of significant digits. This should be 11 ± 2 MHz. Likewise, for other values of other parameters.

Reply: We thank the reviewer for the helpful suggestion. We have corrected the digits in the revised manuscript.

With this corrected, I find the manuscript suitable for publication.

Reply: We thank the reviewer for his/her recommendation for publication.